# Fighting HER2 in Gastric Cancer: Current Approaches and Future Landscapes

**DOI:** 10.3390/ijms26157285

**Published:** 2025-07-28

**Authors:** Margherita Ratti, Chiara Citterio, Elena Orlandi, Stefano Vecchia, Elisa Anselmi, Ilaria Toscani, Martina Rotolo, Massimiliano Salati, Michele Ghidini

**Affiliations:** 1Oncology and Hematology Department, Piacenza General Hospital, Via Taverna 49, 29121 Piacenza, PC, Italy; c.citterio@ausl.pc.it (C.C.); e.orlandi@ausl.pc.it (E.O.); e.anselmi@ausl.pc.it (E.A.); i.toscani@ausl.pc.it (I.T.); 2Pharmacy Unit, Piacenza General Hospital, Via Taverna 49, 29121 Piacenza, PC, Italy; s.vecchia@ausl.pc.it; 3School of Medical Oncology, University Hospital of Modena and Reggio Emilia, Largo del Pozzo 71, 41124 Modena, MO, Italy; martina.rotolo@unimore.it; 4Division of Medical Oncology, University Hospital of Modena, 41124 Modena, MO, Italy; salati.massimiliano@aou.mo.it; 5Oncology Unit, Fondazione IRCCS Ca’ Granda Ospedale Maggiore Policlinico, Via Francesco Sforza 35, 20122 Milan, MI, Italy; michele.ghidini@policlinico.mi.it

**Keywords:** gastric cancer, HER2, HER2-directed strategies

## Abstract

Gastric cancer (GC) remains a major cause of cancer-related mortality worldwide, with human epidermal growth factor receptor 2 (HER2)-positive disease representing a clinically relevant subset. Trastuzumab combined with chemotherapy is the standard first-line treatment in advanced settings, following the landmark ToGA trial. However, resistance to trastuzumab has emerged as a significant limitation, prompting the need for more effective second-line therapies. Trastuzumab deruxtecan, a novel antibody–drug conjugate (ADC) composed of trastuzumab linked to a cytotoxic payload, has demonstrated promising efficacy in trastuzumab-refractory, HER2-positive GC, including cases with heterogeneous HER2 expression. Other HER2-targeted ADCs are also under investigation as potential alternatives. In addition, strategies to overcome resistance include HER2-specific immune-based therapies, such as peptide vaccines and chimeric antigen receptor T cell therapies, as well as antibodies targeting distinct HER2 domains or downstream signaling pathways like PI3K/AKT. These emerging approaches aim to improve efficacy in both HER2-high and HER2-low GC. As HER2-targeted treatments evolve, addressing resistance mechanisms and optimizing therapy for broader patient populations is critical. This review discusses current and emerging HER2-directed strategies in GC, focusing on trastuzumab deruxtecan and beyond, and outlines future directions to improve outcomes for patients with HER2-positive GC across all clinical settings.

## 1. Introduction

Gastric cancer (GC) is the fifth most common malignancy and the fifth leading cause of cancer-related deaths worldwide, with a 5-year survival rate of 20% [1,2]. Most GC patients are diagnosed at advanced stages, which results in poor prognosis due to the lack of typical clinical presentations [3,4]. Traditional treatments for GC mainly include resection, chemotherapy, and radiotherapy [5,6]. Resection is the mainstay treatment for GC, but its application is limited for late-stage GC [7,8,9]. Chemotherapy and radiotherapy are often used before and/or after surgery, but their effectiveness is limited and largely heterogeneous, often accompanied by a range of adverse events [10,11,12]. However, with advances in research on the mechanisms of GC, targeted therapy and immunotherapy have emerged as promising areas of treatment [13]. Targeted therapy specifically identifies and attacks cancer cells by targeting molecular markers, causing less damage to normal cells [14,15]. This therapy offers advantages such as improved therapeutic effects and reduced adverse effects, making it a focal point in personalized treatment strategies for GC. Notably, trastuzumab, combined with chemotherapy, has been shown to increase overall survival (OS) by 2.7 months compared with chemotherapy alone in the ToGA study [16]. However, after the initial progress, the development of effective therapeutic agents for GC stagnated for almost a decade. Although drugs targeting the epidermal growth factor receptor (EGFR), human epidermal growth factor receptor 2 (HER2), and vascular endothelial growth factor (VEGF)/VEGF receptor were also under development during this period, none of them demonstrated significant efficacy.

Recently, however, this stagnant phase has been overcome with the advent of novel anti-HER2 agents and immunotherapies for GC. New anti-HER2 therapeutics, such as trastuzumab deruxtecan ((T-DXd); DS-8201a) and disitamab vedotin (RC48), have shown significant breakthroughs in the treatment of GC [17]. However, it is important to note that while these agents have shown promising mechanisms of action and early efficacy signals, their actual impact on long-term survival outcomes remains limited. In breast cancer, for example, despite the approval of over 20 new agents targeting HER2 since 2000, the Surveillance, Epidemiology, and End Results program of the United States National Cancer Institute data indicate that OS has not substantially improved during this period. This observation calls for a more cautious interpretation of efficacy claims and highlights the importance of robust, survival-based endpoints in clinical trials. Immunotherapy harnesses the body’s own immune system to fight cancer by enhancing immune responses against cancer cells, leading to prolonged survival in GC patients [18]. Specifically, anti-programmed cell death protein 1 (PD-1) antibodies have shown durable anticancer immunity and longer survival in patients with untreated metastatic GC who have microsatellite instability-high or deficient mismatch repair [19]. Emerging immunotherapies, such as adoptive immune cell therapy, tumor vaccines, non-specific immunomodulators, and oncolytic viruses, have expanded the treatment options for GC, providing more personalized and comprehensive approaches. The combination of targeted therapy and immunotherapy is likely to be a future direction of research to improve the efficacy of GC treatments further.

## 2. Classification of Anti-HER 2 Agents

The current classification of HER2-targeted drugs includes monoclonal antibodies (mAbs), tyrosine kinase inhibitors (TKIs), antibody–drug conjugates (ADCs) and bispecific antibodies.

MAbs were the first anti-HER2 targeted therapy, discovered in the 1990s. Their effects are exerted in multiple ways, including HER2 protein downregulation, prevention of HER2-containing heterodimer formation, initiation of G1 cell cycle arrest by induction of the p27 tumor suppressor, prevention of HER2 cleavage, inhibition of angiogenesis, and induction of immune mechanisms (Figure 1) [20].

Trastuzumab and pertuzumab are two specific mAbs that have significantly impacted the treatment landscape of HER2-positive cancers. Trastuzumab, first approved by the Food and Drug Administration (FDA) in 1998, binds to the extracellular domain (ECD) IV of the HER2 receptor, inhibiting downstream signaling pathways and inducing antibody-dependent cell-mediated cytotoxicity (ADCC) [21,22]. Clinical trials with chemotherapy combinations demonstrating improved survival established trastuzumab as the standard-of-care treatment for both metastatic and early-stage HER2-positive breast cancer (BC). Trastuzumab has since gained indications for the treatment of patients with metastatic HER2-positive gastric (GC) and colorectal cancers. Pertuzumab, first approved by the FDA in 2012, binds to ECD II of the HER2 receptor and can act synergically with trastuzumab by preventing the heterodimerization of HER2 with other HER receptors, resulting in further inhibition of downstream tumor signaling [23]. The addition of pertuzumab to trastuzumab has been shown to improve therapeutic benefit by blocking HER2/HER3 signaling [24]. Given their complementary mechanisms of action at the HER receptor level, their effect on immune-mediated antitumor activity, and their complement-mediated cytotoxicity, the combination of these two agents is thought to be synergistic (Figure 2) [25,26].

Margetuximab is the newest mAb targeting HER2, which began to be used in 2020. It binds the identical epitope of HER2 receptor, but it has a much stronger affinity, due to a replacement of five amino acids in the IgG1 Fc domain, which leads to its improved ADCC. This agent is able to maintain trastuzumab antiproliferative effects while also enhancing the activation of innate and adaptive immune responses [27].

TKIs specific to HER2 work by competing with adenosine triphosphate for binding at the HER2 catalytic kinase domain, blocking HER2 signaling. Most of them target more than one HER receptor, with the advantage of simultaneously blocking two or more heterodimer components [28]. This leads to an accumulation of inactive receptors at the cell surface, which also enhances immune-mediated mAb-dependent cytotoxicity [29]. Some advantages of TKIs are their small molecular size, their oral bioavailability, and their ability to cross the blood–brain barrier. The first anti-HER2 TKI was lapatinib, a reversible TKI first FDA-approved in 2007 in combination with capecitabine for patients who progressed on mAb-based therapy. Three years later, lapatinib was approved with the use of letrozole as a first-line therapeutic option for triple-positive BC [30]. Neratinib is a second-generation, irreversible pan-HER TKI that targets EGFR, HER2, and HER4, which leads to a greater effect than lapatinib but also higher toxicity, especially diarrhea. This agent was first FDA-approved in 2017 for extended adjuvant treatment in early-stage HER2-positive BC [31]. Tucatinib is a third-generation, reversible, highly selective anti-HER2 TKI that has >1000-fold greater potency for HER2 than EGFR. Diarrhea, nausea, hand–foot syndrome, and fatigue are the most common adverse effects. It has also demonstrated high levels of penetration in the central nervous system (CNS). It was first FDA-approved in 2020 for the treatment of HER2-positive metastatic BC, including in patients with CNS metastases [32].

An innovative drug family designed to target HER2 is represented by ADC. ADC is composed of an antibody (usually humanized or chimeric immunoglobulin G), a linker, and a cytotoxic payload, a highly potent agent delivered directly into the tumor cells via antibody-mediated endocytosis. The linker binds the payload to the antibody and must be stable in circulation to deliver the payload directly to the tumor cells and avoid its premature release in the bloodstream [33]. After binding of the ADC–HER2 complex, it is internalized in an endosome and transported to lysosomes, so the released payload can elicit antitumor activity within targeted cells. The payload can be released within the extracellular space before or after the ADC internalization, where it can also exert its activity in the neighboring cells, which may or may not express HER2. This action, known as the “bystander effect”, has markedly improved the activity of ADCs in cancer with heterogenous and/or low HER2 expression [33]. Trastuzumab-emtansine (TDM-1), a second-generation ADC, was first approved by the FDA in 2013 for HER2-positive metastatic BC (in second- and third-line settings, following the failure of trastuzumab and taxane), and subsequently, in 2019 it was approved in the adjuvant setting for early-stage HER2-positive BC with residual invasive disease after neoadjuvant treatment [34]. The most common adverse effects are fatigue, thrombocytopenia, increased aminotransferase levels, and neuropathy. T-DXd, a third-generation ADC with a potent bystander effect, was demonstrated to be significantly superior to TDM-1 in previously treated HER2-positive metastatic BC [35]. T-DXd was first approved by the FDA for this indication in 2019, and it has since also been approved for HER2-low metastatic BC, HER2-positive advanced GC, and HER2-mutant metastatic non-small-cell lung cancer. The most common reported adverse effects include neutropenia, anemia, and nausea. It was also associated with an increased risk of interstitial lung disease.

Bispecific antibodies (BsAbs) represent another class of drugs designed to simultaneously bind to two different antigens, which can enhance therapeutic efficacy. Zanidatamab (ZW25) is a novel HER2-targeting BsAb that binds to HER2 ECD II and IV, the same domains targeted by trastuzumab and pertuzumab. According to a phase I trial, ZW25 was well tolerated and showed durable anticancer activity in heavily pretreated gastroesophageal adenocarcinoma patients [36]. First-line ZW25 plus chemotherapy in HER2-expressing gastroesophageal adenocarcinoma also showed a good tolerability, manageable safety profile, and durable response in a multicenter phase II study [37]. Based on these findings, a global phase III trial was designed to evaluate the efficacy and safety of ZW25 combination chemotherapy with or without tislelizumab compared with standard therapy (trastuzumab plus chemotherapy) as first-line treatment of patients with metastatic HER2-positive gastroesophageal adenocarcinoma [38]. KN026 is another anti-HER2 BsAb that binds non-overlapping epitopes of HER2, leading to dual HER2 signaling blockade and achieving the effect of trastuzumab plus pertuzumab. A phase II trial (NCT03925974) showed that patients with HER2-expressing advanced GC or GEJA receiving KN026 had an objective response rate (ORR) of 56% and a durable remission duration of 9.7 months [39].

## 3. Targeting HER2 in Perioperative and Neoadjuvant Settings

The addition of perioperative chemotherapy to curative intent surgery has long been recognized as the Western standard of care for resectable stage IB-III GC based on an increase in OS and disease-free survival (DFS) compared with surgery alone [40,41]. The FLOT regimen (5-fluorouracil, leucovorin, oxaliplatin, and docetaxel) outperformed the antracycline containing triplet ECF/ECX (epirubicin, cisplatin, 5-fluorouracil/ capecitabine) in the randomized phase II-III FLOT4 trial, showing better outcomes without significant impact on toxicity, thus becoming the preferred perioperative option for patients fit for a three-drug regimen [42,43].

In resectable GC, HER2 overexpression/amplification is found in approximately 10–20% of cases [44,45], and it is associated with a proximal subsite, intestinal histologic type, lymph nodes metastasis, and more advanced disease stage [46]. The prognostic role of HER2 overexpression and/or amplification is controversial in GC, with series suggesting worse survival for patients with HER2-positive disease [47,48] and others reporting no prognostic value based on different HER2 expression [49]. Regarding its potential as a predictive biomarker, HER2 status was not associated with enhanced benefit from chemotherapy in a translational analysis involving 415 patients from the MAGIC trial [50].

The positive results observed in advanced GC [16], together with data in early-stage breast cancer [51], have fostered the development of HER2-directed therapies in the resectable disease setting (Table 1). Two single-arm phase II trials have investigated the addition of the anti-HER2 mAb trastuzumab to perioperative doublet and triplet chemotherapy in GC. The Spanish NEOHX trial included 36 patients treated in the neoadjuvant setting with capecitabine, oxaliplatin (XELOX) and trastuzumab followed by surgery and subsequent postoperative XELOX–trastuzumab and maintenance trastuzumab for 12 cycles [52]. After three cycles of preoperative XELOX–trastuzumab, no disease progression was recorded, 14 patients (38%) had partial response (PR), 18 (50%) had SD, and 28 (90%) had R0 resection. A pathologic complete response was observed in three patients (9.6%). More importantly, the study was statistically positive, demonstrating an 18-month DFS of 71%. Likewise, the German HER-FLOT trial was conducted in 56 patients with cT2–T4 cN + HER2-positive disease evaluating the standard FLOT regimen (i.e., four cycles preoperative and four cycles postoperative) in combination with trastuzumab, administered concomitantly to chemotherapy (4 mg/kg bi-weekly) and then sequentially (6 mg/kg three-weekly) [53]. Akin to the adjuvant treatment of breast cancer, trastuzumab therapy was foreseen for a total of 12 months. The study met its primary endpoint, overcoming the 20% cut-off rate of centrally confirmed pCR (i.e., 21.4%; 12 out of 56 patients), reporting one of the highest rates ever shown in a prospective trial in this setting. Of note, nearly half of patients achieved a complete or near-complete remission (46.4%). In addition, the R0 resection rate was 92.9% with no unexpected safety issues. In terms of survival outcomes, the median DFS and the 3-year OS were a promising 42.5 months and 82.1%, respectively. In recent years, another trial has reported results for the dual blockade of HER2 with both trastuzumab and pertuzumab associated with FLOT chemotherapy compared with FLOT alone [54]. Despite stopping prematurely due to the negative results of the JACOB trial, the AIO PETRARCA trial showed a significantly improved pCR rate for the addition of trastuzumab and pertuzumab to FLOT (35% vs. 12%, *p* = 0.019) after the inclusion of 80 patients during the randomized phase II part.

Taking into account that most of HER2-positive GC originates from the gastroesophageal junction (GEJ) and that these tumors have been historically treated with neoadjuvant chemoradiotherapy as an alternative to perioperative chemotherapy, the targeting of HER2 has been investigated also in combination with the CROSS regimen. In this regard, the TRAP study reported good safety, tolerability, and pCR in 13 out of 40 patients (34%) for the addition of trastuzumab and pertuzumab to standard chemoradiotherapy in adenocarcinoma of the esophagus or GEJ [55]. Notably, with a median follow-up of 32 months, 3-year OS was 71%, comparing favorably with historical controls. Contrariwise, the combination of trastuzumab with trimodality treatment (CROSS regimen followed by surgery) neither improved the pCR rate nor DFS in a similar patient population in the randomized phase III RTOG 1010 trial [56].

Collectively, the above-reported results demonstrated the feasibility, good tolerability, and promising antitumor activity for HER2-directed agents combined with chemotherapy as a perioperative strategy in molecularly selected GC. However, data from well-powered and properly conducted randomized trials are needed to assess the potential impact on survival of this multimodal approach.

To this end, the results of the open-label phase 2 EORTC INNOVATION trial (NCT02205047) have recently been presented at the 2025 American Society of Clinical Oncology Gastrointestinal Cancer Symposium [57]. Overall, 161 patients with stage I–III, centrally determined HER2 overexpressing GC and GEJ tumors were randomly allocated 1:1:2 into three perioperative treatment arms: chemotherapy alone (arm A), chemotherapy plus trastuzumab (arm B), or chemotherapy in combination with trastuzumab and pertuzumab (arm C). The chemotherapy backbone per investigator’s choice included 5-fluorouracil (5-FU) single-agent, cisplatin/5-FU, XELOX, and FOLFOX (5-fluorouracil, leucovorin, oxaliplatin), and after the publication of FLOT-4 in 2019, the FLOT regimen was incorporated in European sites. Notably, when stratified by a chemotherapy backbone, the data showed a clear improvement in major pathologic response rate for FLOT or XELOX/mFOLFOX6 plus trastuzumab (53.3%) compared with FLOT or XELOX/mFOLFOX6 plus dual HER2 blockade (37.9%) and the same chemotherapy regimens alone (33.3%). In terms of survival, while a non-significant advantage in both 3-year RFS and 3-year OS was evident in the trastuzumab plus chemotherapy arm before the protocol amendment (HR 0.64 and 0.77, respectively), after that, the benefit was no longer observed (HR 1.12 and 0.99, respectively). Importantly, the combination of chemotherapy with a dual HER2 blockade resulted in detrimental outcomes. The higher toxicity (especially diarrhea and neutropenia) and lower dose intensity recorded in arm C have been advocated to explain these results. Acknowledging major limitations of this study, including the small sample size in each arm, the short follow-up (median 4.3 years), and the mid-study amendment changing in chemotherapy regimen, a potential role for trastuzumab cannot be excluded in the future, particularly in patients requiring downsizing.

Despite these disappointing data, further research efforts are underway to improve the outcome of HER-positive GC patients. Strategy incorporating newer-generation anti-HER2 agents such as the ADC T-DXd [58] or immune checkpoint inhibitors [59] within the perioperative regimen are under active investigation in this disease setting. The results of these ongoing trials are eagerly awaited to refine the standard-of-care management of HER2-positive resectable GC.

## 4. Targeting HER2 in Advanced Disease

Among the many clinical trials assessing HER2-targeted strategies in advanced GC, only a few—namely the ToGA trial, KEYNOTE-811, and DESTINY-Gastric01/02—have demonstrated statistically and clinically meaningful results. Other studies, while contributing valuable insights, have shown limited or inconclusive benefit, reinforcing the need for further research and more consistent outcomes in broader populations.

Starting from the ToGA trial, this the first randomized controlled study to establish trastuzumab plus chemotherapy (cisplatin and either 5-FU or capecitabine) as the first-line standard of care for HER2-positive advanced gastric or GAJ adenocarcinoma. In 594 patients, trastuzumab significantly improved median OS from 11.1 to 13.8 months (HR 0.74; *p* = 0.0046), with the most pronounced benefit observed in patients with HER2 IHC 3 + or IHC 2 + /ISH + tumors with ≥6 gene copies (HR 0.65 and median OS of 16.0 months versus 11.8 months in the chemotherapy group) [16]. There was evidence of a significant interaction test (*p* = 0.036) between treatment and the two HER2 subgroups (high HER2 expression vs. low HER2 expression). The toxicity profile was comparable between groups, with a slight increase in diarrhea and neutropenia in the trastuzumab arm [16]. Cardiac adverse events were rare and similar between groups, with serious grade 3–4 events being slightly more frequent in the chemotherapy-alone arm (3% vs. 1%), while clinically significant cardiac dysfunction (≥10% LVEF drop to <50%) occurred in 5% of patients receiving trastuzumab plus chemotherapy versus 1% with chemotherapy alone [16].

Following the ToGA study, further research aimed to enhance anti-HER2 efficacy by combining trastuzumab with other agents.

The phase III JACOB trial investigated whether adding pertuzumab, a mAb that inhibits HER2/HER3 dimerization, to trastuzumab and chemotherapy could improve outcomes in the first-line setting. Median OS increased from 14.2 to 17.5 months (HR 0.84), but the difference was not statistically significant (*p* = 0.057) [2]. Diarrhea (13% grade 3–4) and neutropenia (30%) were more common in the pertuzumab group [2]. The failure of the JACOB study to show a significant benefit may be partly explained by the marked heterogeneity of HER2 expression in GC, both spatially, with amplification-positive cells ranging from 10% to nearly 100% [60] and intratumoral heterogeneity being observed in over 40% of cases; and temporally, with HER2 loss occurring in 30–70% of patients after trastuzumab-based therapy [61]. Additionally, variability in HER2 immunohistochemical staining between tumor samples further complicates accurate classification and treatment targeting.

The phase III KEYNOTE-811 trial evaluated the addition of pembrolizumab to trastuzumab and chemotherapy in HER2-positive metastatic gastric or GEJ cancer. An interim analysis showed a significantly higher ORR with pembrolizumab (74.4% vs. 51.9%; *p* = 0.00006), including an 11% complete response rate. Immune-related adverse events, including thyroiditis, hepatitis, and pneumonitis, were more frequent in the experimental arm [37]. The combination appeared effective regardless of programmed death ligand 1 (PD-L1) expression status, although progression-free survival (PFS) and OS benefits were more evident in PD-L1 CPS ≥ 1 subgroups [37]. Based on these findings, the FDA and European Medicines Agency (EMA) approved the regimen of pembrolizumab, trastuzumab, and chemotherapy for the first-line treatment of HER2-positive GC. At the final analysis (median follow-up 50.2 months), the combination significantly improved OS compared with placebo (20.0 vs. 16.8 months; HR 0.80; *p* = 0.0040), with even greater benefit in PD-L1 CPS ≥ 1 patients (HR 0.79) [8]. PFS and ORR remained superior in the pembrolizumab arm, confirming this triplet as a new standard of care in this setting.

The phase II randomized INTEGA trial compared two experimental first-line regimens in HER2-positive metastatic gastric and GEA: trastuzumab plus nivolumab combined either with ipilimumab (chemo-free arm) or with FOLFOX (chemo-based arm) [62]. Although the ipilimumab arm showed manageable toxicity, it did not improve outcomes, with lower 12-month OS (57% vs. 70%), shorter median PFS (3.2 vs. 10.7 months), and reduced ORR (32% vs. 56%) compared with the FOLFOX arm. Median OS was comparable between groups (16.4 vs. 21.8 months), but survival curves crossed over time, suggesting that a subset may benefit from chemo-free immunotherapy. These findings appear to confirm the central role of chemotherapy in the first-line treatment of HER2-positive esophagogastric adenocarcinoma and support the added value of immunotherapy when used in combination. Translational analyses in the INTEGA trial confirmed substantial molecular heterogeneity, with baseline tumor tissue and cfDNA profiling revealing frequent driver and resistance-associated mutations—most commonly in TP53, ERBB2, and PIK3CA—and an early increase in cfDNA levels (>20% after one cycle), strongly predicting shorter PFS and OS, regardless of treatment arm. Moreover, on-treatment liquid biopsies identified acquired ERBB2 resistance mutations, including truncating variants causing loss of membrane HER2 expression and a point mutation (H574L) within the trastuzumab-binding domain, indicating strong therapeutic pressure and the emergence of trastuzumab escape mechanisms [62].

In addition to checkpoint inhibitors, other investigational strategies include bispecific antibodies such as zanidatamab, which binds two distinct HER2 epitopes. In a recent phase II study in first-line treatment of HER2-positive metastatic GEA, zanidatamab combined with chemotherapy demonstrated a confirmed ORR of 84% among centrally confirmed HER2-positive patients, including four complete responses [63]. The median duration of response was 18.7 months, and median PFS reached 15.2 months. Although median OS was not yet mature, 24- and 30-month OS rates were 65% and 59%, respectively. The combination was generally well tolerated, with diarrhea (35%) and hypokalemia (22%) being the most common grade ≥3 treatment-related adverse events and prophylactic antidiarrheal measures substantially reducing the rate of severe diarrhea [63].

The ADC T-DXd are being investigated in the first-line setting in combination with chemotherapy and immunotherapy. Preliminary results from the DESTINY-Gastric03 study, which is evaluating first-line combinations involving T-DXd, are promising. In part 2 of the trial, patients treated with T-DXd in combination with fluoropyrimidine and pembrolizumab demonstrated an ORR of 78%, highlighting substantial antitumor activity. Although mature data are still awaited, these early results underscore the potential role of ADC in the frontline setting [64].

In the second-line setting, the GATSBY phase II/III trial compared trastuzumab emtansine with taxanes (docetaxel or paclitaxel) in patients previously treated with trastuzumab-based therapy [65]. T-DM1 did not improve OS (7.9 vs. 8.6 months; HR 1.15; *p* = 0.86), though it showed a lower incidence of grade ≥3 neutropenia but higher rates of anemia (26%) and thrombocytopenia (11%). One explanation for T-DM1 underperformance lies in GC’s intrinsic heterogeneity and frequent HER2 loss after first-line therapy, which reduces the efficacy of HER2-targeted agents [60,61,66]. Furthermore, mechanisms like MMP7 upregulation and the DKK1–Wnt/β-catenin pathway have been implicated in resistance to T-DM1 [66,67].

T-DXd demonstrated significant benefits in two pivotal trials that led to its approval by both the EMA and FDA for second-line treatment of HER2-positive advanced gastric or GEJ cancer [4,14]. DESTINY-Gastric01, a randomized phase II trial in Japan and South Korea, enrolled patients with HER2-positive advanced gastric or GEJ adenocarcinoma who had received ≥2 prior lines of therapy, including trastuzumab. T-DXd improved ORR (51% vs. 14%) and median OS (12.5 vs. 8.4 months; HR 0.59; *p* = 0.01) compared with physician’s choice chemotherapy. Interstitial lung disease occurred in 10% of patients, requiring monitoring [4]. DESTINY-Gastric02, a single-arm phase II study conducted in the U.S. and Europe, included patients who progressed after first-line trastuzumab-based therapy. In this Western population, T-DXd achieved an ORR of 38% (95% CI: 27.3–49.6), median PFS of 5.6 months, and median OS of 12.1 months. Interstitial lung disease occurred in 6.3% of patients, all grade 1–2, confirming a manageable safety profile [14]. Together, these data support the use of T-DXd as a new standard in the second-line setting, based on both its demonstrated efficacy and manageable safety profile, particularly in patients previously treated with trastuzumab.

## 5. Future Perspectives

Chimeric antigen receptor T (CAR-T) cells represent a novel form of precision-targeted immunotherapy, involving genetically modified patient- or donor-derived T cells engineered to recognize tumor-associated antigens [68]. HER2-specific CAR-T cells have demonstrated superior tumor suppression, extended survival, and enhanced tumor targeting in HER2-positive xenograft models, outperforming non-transduced T cells [69].

A phase I clinical trial involving eleven patients with HER2-positive advanced biliary tract and pancreatic cancers (excluding GC) reported a PR lasting 4.5 months in one case and stable disease (SD) in five others. The median PFS was 4.8 months (range: 1.5–8.3 months) [70]. Similarly, a recent first-in-human phase I trial assessed the safety profile of CT-0508, an autologous macrophage-based therapy engineered to express an anti-HER2 CAR, in HER2-positive tumors. Although gastric or esophageal adenocarcinoma patients were not included, one patient with primary cholangiocarcinoma and gastric metastases experienced a transient, severe upper gastrointestinal hemorrhage 11 days post-infusion [71].

An alternative approach with potentially reduced toxicity is the T cell antigen coupler platform, which utilizes a TAC receptor to redirect T cells while co-opting the native T cell receptor (TCR), resulting in more controlled immune responses than CAR-T cells. In a phase I trial evaluating the safety and efficacy of autologous TAC01-HER2 in refractory HER2-expressing solid tumors, a PR was observed in a pretreated GC patient [72].

Beyond CAR-T cells, HER2-targeted dendritic cell vaccines have also been explored. One study enrolled nine GC patients who received four intradermal HER2 peptide-pulsed dendritic cell vaccinations at biweekly intervals. Among these, one patient achieved a PR alongside a decline in tumor markers, while another maintained SD for three months [73]. A separate phase I study using DCs transduced with an adenoviral vector encoding HER2′s extracellular and transmembrane domains recorded a PR in one GC patient and prolonged SD (24 weeks) in another pretreated with anti-HER2 therapy [74]. Additionally, the B-VAC vaccine—an autologous B cell- and monocyte-derived HER2-targeting vaccine—was tested in a first-in-human phase I trial. Although six patients received high-dose formulations, only three achieved SD, with no objective being responses observed [75].

The HERIZON phase II trial, a multicenter, randomized, open-label study, evaluated HER-Vaxx (IMU-131), a B cell peptide vaccine targeting HER2, in combination with chemotherapy versus chemotherapy alone in treatment-naïve patients with GC. The experimental arm demonstrated a 40% improvement in OS (HR 0.60; median OS 13.9 months; 80% CI: 7.52–14.32) versus 8.31 months in the control arm (80% CI: 6.01–9.59). PFS was also improved by 20% (HR 0.80; 80% CI: 0.47–1.38). Vaccination was well tolerated and induced robust HER2-specific IgG and IgG1 antibody responses (*p* < 0.001 vs. controls), which significantly correlated with tumor reduction (IgG: *p* = 0.001; IgG1: *p* = 0.016) [76].

Immune-stimulating antibody conjugates (ISACs) have also been evaluated in HER2-overexpressing tumors, including GC. ISACs consist of antibodies conjugated to toll-like receptor agonists to induce a tumor-specific adaptive immune response. In a phase I/II first-in-human trial, BDC-1001—an ISAC composed of a trastuzumab biosimilar linked to a toll-like receptor 7/8 agonist—was tested alone or with nivolumab. Four PRs were reported across a mixed HER2 expression cohort, although none were GC cases. Ten patients experienced SD lasting ≥6 months, including individuals with GC [77].

Unlike breast cancer, the therapeutic potential of targeting HER2-low GC remains controversial and is still under investigation [78]. ADCs have emerged as a promising strategy for addressing HER2-low GC. The DESTINY-Gastric03 trial (phase Ib/II) is currently assessing T-DXd in combination with fluoropyrimidine and volrustomig, a bispecific PD-1/CTLA-4 antibody, in HER2-low GC (arm 3B) [79]. Another ADC, SHR-A1811—comprising an anti-HER2 IgG1 antibody linked to the topoisomerase I inhibitor SHR9265—has shown potent cytotoxicity in HER2-high and HER2-null cell lines. In a phase I trial, SHR-A1811 induced tumor regression in 12 HER2-low GC patients, including four with PRs [80].

Disitamab vedotin (RC48), an ADC incorporating hertuzumab (a novel anti-HER2 mAb) and monomethyl auristatin E (MMAE), exerts both direct cytotoxic and immune stimulatory effects via the STING pathway [79]. A phase I trial of RC48 combined with toripalimab in HER2-expressing advanced GC showed an overall response rate (ORR) of 43% (12/28), including a 46% ORR in HER2-low cases (5/9 patients) [81].

XMT-2056, a STING agonist ADC targeting a novel HER2 epitope distinct from trastuzumab and pertuzumab, has shown encouraging preclinical activity in GC models with varying HER2 expression, especially when combined with other HER2-targeted therapies. The compound has received orphan drug designation from the FDA [82,83].

Cinrebafusp alfa (PRS-343), a bispecific fusion protein combining HER2-binding domains and the T cell costimulatory receptor CD137, has shown favorable tolerability and preliminary antitumor activity in HER2-positive solid tumors. A phase II multi-cohort trial is underway to assess its efficacy in combination with tucatinib in HER2-low advanced GC and tumors harboring HER2-activating mutations [84,85].

Lastly, HF-K1, a trastuzumab Fab-conjugated immunoliposome encapsulating doxorubicin, represents a novel HER2-targeting strategy under investigation for both HER2-positive and HER2-negative tumors. Preclinical models have demonstrated activity in HER2-low xenografts, and a phase I clinical trial is currently ongoing [86].

## 6. Conclusions

The combination of trastuzumab and platinum-based chemotherapy tested in ToGA trial has established itself as the standard-of-care treatment for HER2-positive GC, significantly improving survival outcomes for patients with advanced disease [16]. However, despite these advancements, drug resistance remains a substantial challenge, particularly after the failure of trastuzumab-based first-line therapy. While other HER2-targeted agents, such as lapatinib, pertuzumab, and T-DM1, have been evaluated in clinical trials, they have not demonstrated substantial improvements in patient outcomes beyond the first-line setting [87]. This underscores the critical need for novel therapeutic strategies to overcome resistance mechanisms and optimize long-term survival in HER2-positive GC patients.

The emergence of ADC has reinvigorated the therapeutic landscape, offering a promising avenue for overcoming the limitations of traditional HER2-targeted therapies [88]. T-DXd, a next-generation ADC, has shown notable clinical activity, including improvements in ORR and OS in heavily pretreated HER2-positive GC patients, as demonstrated in DESTINY-Gastric01 and DESTINY-Gastric02 trials [89,90,91]. Its unique mechanism, which utilizes a bystander effect to target tumor cells irrespective of HER-2 expression levels, is particularly advantageous in GC, a disease frequently characterized by heterogeneous HER2 expression [92].

Ongoing clinical trials are currently evaluating T-DXd in earlier disease stages, including as a first-line treatment, with the aim of extending its therapeutic benefit to a broader patient population [93]. In parallel, other ADCs, such as disitamab vedotin, have shown promising preliminary results in certain populations, such as Asian patients [94]. However, further large-scale multinational trials are necessary to validate the applicability of these agents.

In addition to ADCs, bispecific antibodies represent a promising class of therapeutics in the treatment of HER2-positive GC. Margetuximab, a mAb with enhanced binding to the Fc receptor, and zanidatamab, a bispecific antibody targeting both HER2 and CD3, have demonstrated encouraging preclinical and early clinical signals, particularly in first-line settings, although confirmatory evidence from large randomized trials is still awaited [95]. These agents are designed to augment immune system-mediated tumor cell destruction by engaging multiple immune pathways simultaneously, potentially offering improved efficacy over traditional mAbs.

Furthermore, immune checkpoint inhibitors, particularly those targeting the PD-1/PD-L1 axis, are being investigated in combination with HER2-targeted therapies. The results of the KEYNOTE-811 trial, which demonstrated the clinical benefit of combining trastuzumab with pembrolizumab (a PD-1 inhibitor) in HER2-positive, PD-L1-positive GC, may represent a significant shift in the treatment paradigm, especially when combined with immunotherapy in selected patient subgroups [96].

Chimeric antigen receptor T (CAR-T) cell therapy, although primarily a breakthrough in the treatment of hematologic malignancies, is being explored for its potential application in HER2-positive GC [97]. However, the translation of CAR-T cell therapy to solid tumors, including GC, remains fraught with challenges. Tumor heterogeneity, the immunosuppressive tumor microenvironment, and the potential for antigen escape mechanisms complicate the ability of CAR-T cells to consistently target and eradicate tumor cells [95,98]. Moreover, the risk of severe immune-related adverse events, such as cytokine release syndrome, further underscores the need for careful optimization of CAR-T cell therapy in this setting. Despite these challenges, ongoing research focused on enhancing CAR-T cell specificity, overcoming immune evasion, and modulating the tumor microenvironment holds the potential—pending resolution of current technical and safety challenges—to make CAR-T cell therapy a viable treatment option [95].

The advent of liquid biopsy technologies also offers significant promise for improving the management of HER2-positive GC [87]. Liquid biopsy enables non-invasive, real-time monitoring of disease progression and molecular profiling, facilitating earlier detection of minimal residual disease and enabling dynamic adjustments to treatment regimens. The ability to identify acquired resistance mutations and track changes in HER2 expression during treatment provides valuable insights that can guide clinical decision-making, optimizing therapy for individual patients.

Looking to the future, the treatment of HER2-positive GC is poised to benefit from several exciting advancements [99]. The identification of novel predictive biomarkers will be crucial in selecting patients most likely to benefit from HER2-targeted therapies, ensuring that treatments are tailored to the individual molecular and genetic characteristics of each patient [100]. Furthermore, the continued evolution of personalized medicine, incorporating factors such as genetic background, tumor heterogeneity, and immune status, is likely to lead to a more refined and individualized approach to treatment [87].

In conclusion, while significant progress has been made in the treatment of HER2 GC, the challenges of drug resistance, the limited efficacy of existing therapies after trastuzumab failure, and the need for better predictive biomarkers remain formidable hurdles. However, with the development of novel therapeutic modalities, including advanced ADCs, bispecific antibodies, immune checkpoint inhibitors, CAR-T cells, and liquid biopsy, the treatment landscape for HER2-positive GC is poised for transformative advancements. Ongoing clinical research and collaborative efforts will be essential to refine these emerging strategies and provide more effective, personalized treatment options for patients with this challenging malignancy.

## Figures and Tables

**Figure 1 ijms-26-07285-f001:**
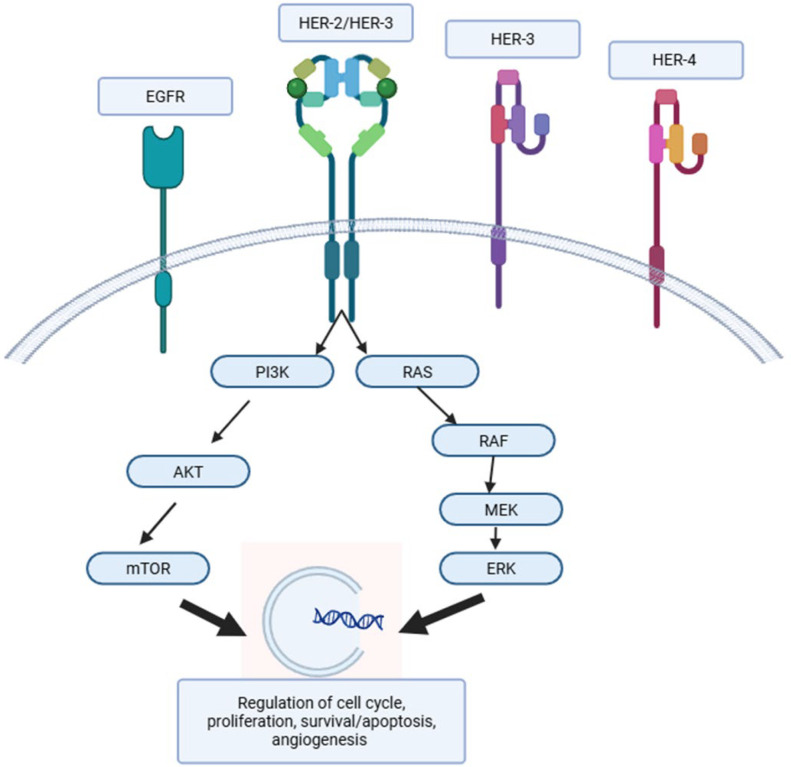
HER2 pathway and its effect on cell cycle regulation.

**Figure 2 ijms-26-07285-f002:**
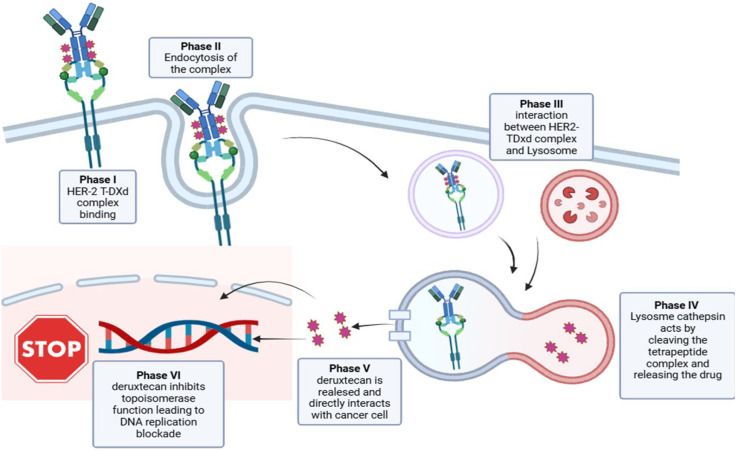
Mechanism of action of trastuzumab deruxtecan and its direct effect on the cancer cell.

**Table 1 ijms-26-07285-t001:** Major clinical trials investigating HER2-directed therapies in the perioperative/neoadjuvant management of resectable GC (CROSS: regimen consisted of weekly paclitaxel and carboplatin concomitant with radiotherapy; DFS, disease-free survival; EA, esophageal adenocarcinoma; FLOT, 5-fluorouracil, oxaliplatin, docetaxel; GA, gastric adenocarcinoma; GEJA, gastroesophageal junction adenocarcinoma; G3-G4 AEs, grade 3 and grade 4 adverse events; OS, overall survival; P, pertuzumab; pCR, pathologic complete response; PFS, progression-free survival; T, trastuzumab; XELOX, capecitabine, oxaliplatin.)

Study Name	Study DESIGN	Treatment Arms	Sample Size	Primary Site	pCR	R0 Resection	DFS	OS	G3–G4 AEs
NEOHX [52]	Phase II, single-arm	XELOX + T	36 pts	GA, GEJA	9.6%	90%	18-month DFS: 71%	mOS:79.9 5-yr OS: 58%	Diarrhea (33%), nausea, and vomiting (8%)
HER-FLOT [53]	Phase II, single-arm	FLOT + T	56 pts	GA, GEJA	21.4%	92.9%	mDFS: 42.5 months	3-yr OS: 82.1	Neutropenia (46.6%), diarrhea (17.0%)
PETRARCA [54]	Phase II/III, randomized	FLOT + T + P	80 pts	GA, GEJA	35%	93%	2-yr DFS: 70%	2-yr OS: 84%	Diarrhea (41%), leukopenia (23%)
TRAP [55]	Phase II, single-arm	CROSS + T + P	40 pts	EA, GEJA	34%	100%	3-yr PFS: 57%	3-yr OS: 71%	Diarrhea (20%), dysphagia (18%)
RTOG-1010 [56]	Phase III, randomized	CROSS + T + P	203 pts	EA, GEJA	27%	98%	4-yr DFS: 33.2%	4-yr OS: 47.6%	Hematological (56%), gastrointestinal disorders
INNOVATION [57]	Phase II, randomized	A: Chemo ° B: Chemo + T C: Chemo + T + P	161 pts A: 33 pts B: 64 pts C: 64 pts	GA, GEJA	A: 33.3% °°″ B: 53.3% C: 37.9%	A: 83.9% B: 90.3% C: 85.9%	3-yr PFS °° A: 68.4% B: 60.5% C: 50.4%	3-yr OS °° A: 73.3% B: 72.2% C: 62.2%	Diarrhea (A: 5.9%, B: 3.0%, C: 26.1%), Neutropenia (A: 32.4%, B: 21.2%, C: 21.7%).

° Chemotherapy included: 5-FU single-agent, cisplatin/5-FU, XELOX, FOLFOX, and after the publication of FLOT-4 in 2019, the FLOT regimen was incorporated in European sites. °° After protocol amendment. ″ These percentages refer to the major pathologic response rate.

## Data Availability

Dataset available on request from the authors.

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
