# Peer review of "Fighting HER2 in Gastric Cancer: Current Approaches and Future Landscapes"

_ijms, 2025, doi:10.3390/ijms26157285_

Round 1
Reviewer 1 Report
Comments and Suggestions for Authors
The following comments might help to improve the manuscript by a more critical attitude towards clinical results and FDA approval.
In section 2, overviewing anti-HER2 agents , mechanisms of action are well described and appropriately illustrated. The authors mention the year of FDA-approval mostly for breast cancer, but too little attention is paid to the evidence (or lack of it) that these new agents are favoring the outcome in terms of survival or other endpoints. This evidence needs a critical approach since survival from breast cancer has not improved since the year 2000 (SEER data), despite the approval of more than 20 new drugs.
In sections 3 and 4, the approach to clinical trials for HER2-postive GC is more in detail, concluding that further research efforts are needed. Out of nine trials , only ToGA, Phase III Keynote-811, and DESTINY-Gastric01 and 02 have generated satisfactory results pointing to significant benefits of HER2-targetting in GC.
Conclusions. In view of the results discussed in the present review, the terms “compelling efficacy” ( L469), “encouraging preclinical and clinical data” (L483), “marked a significant shift” (L491), “holds the promise” (L 503), and “significant progress” (L519) need clarification (criteria?).
Author Response
Comment 1: In section 2, overviewing anti-HER2 agents , mechanisms of action are well described and appropriately illustrated. The authors mention the year of FDA-approval mostly for breast cancer, but too little attention is paid to the evidence (or lack of it) that these new agents are favoring the outcome in terms of survival or other endpoints. This evidence needs a critical approach since survival from breast cancer has not improved since the year 2000 (SEER data), despite the approval of more than 20 new drugs.
Response 1: We thank the reviewer for this insightful comment. In response, we have revised Section 2 to include a more critical discussion regarding the clinical benefit of anti-HER2 agents, particularly in terms of overall survival and other meaningful endpoints. We agree that while many drugs have received FDA approval, their impact on long-term survival in breast cancer remains debated. To provide a balanced perspective, we now refer to SEER data and highlight the limited survival gain observed over time, despite the introduction of multiple agents. This context has been explicitly added to reinforce the need for cautious optimism and the importance of robust endpoints in assessing clinical utility, especially when extrapolating data from breast cancer to gastric cancer.
Original text : "...have demonstrated significant breakthroughs in the treatment of GC [17]."
We modified the text (see the red part in the new version of the manuscript attached), according to the reviewer’ s comment as follow :
" However, it is important to note that while these agents have shown promising mechanisms of action and early efficacy signals, their actual impact on long-term survival outcomes remains limited. In breast cancer, for example, despite the approval of over 20 new agents targeting HER2 since 2000, SEER data indicate that overall survival has not substantially improved during this period. This observation calls for a more cautious interpretation of efficacy claims and highlights the importance of robust, survival-based endpoints in clinical trials. "
Comment 2: In sections 3 and 4, the approach to clinical trials for HER2-postive GC is more in detail, concluding that further research efforts are needed. Out of nine trials , only ToGA, Phase III Keynote-811, and DESTINY-Gastric01 and 02 have generated satisfactory results pointing to significant benefits of HER2-targetting in GC.
Answer: We appreciate the reviewer’s observation and agree that, to date, only a few pivotal trials have shown meaningful clinical benefit in HER2-positive gastric cancer. We have clarified this in the manuscript by emphasizing the modest number of studies with robust results, namely ToGA, KEYNOTE-811, DESTINY-Gastric01, and DESTINY-Gastric02. The remaining trials, while informative, often lack statistical power or yield inconclusive outcomes, reinforcing the need for more rigorous, randomized studies. This acknowledgment has been added to the discussion to avoid overgeneralization of efficacy across all HER2-directed trials in gastric cancer. We modified the manuscript (see the red comments) as follow:
Original text:
"The ToGA trial was the first randomized controlled study..."
*Added in red:*Among the many clinical trials assessing HER2-targeted strategies in advanced gastric cancer, only a few—namely the ToGA trial, KEYNOTE-811, and DESTINY-Gastric01/02—have demonstrated statistically and clinically meaningful results. Other studies, while contributing valuable insights, have shown limited or inconclusive benefit, reinforcing the need for further research and more consistent outcomes in broader populations. Starting from TOGA trial…"
comment 3: Conclusions. In view of the results discussed in the present review, the terms “compelling efficacy” ( L469), “encouraging preclinical and clinical data” (L483), “marked a significant shift” (L491), “holds the promise” (L 503), and “significant progress” (L519) need clarification (criteria?).
Answer to comment for "conclusions" paragraph: Thank you for highlighting the need for more precise language in our conclusions. We have reviewed the use of the terms “compelling efficacy,” “encouraging preclinical and clinical data,” “marked a significant shift,” “holds the promise,” and “significant progress.” Where appropriate, we have revised the text to either qualify these expressions with specific clinical endpoints (e.g., OS improvement, ORR, pCR rates) or to adopt a more cautious tone. For instance, “compelling efficacy” in reference to trastuzumab deruxtecan is now supported by ORR and OS data from DESTINY-Gastric01/02, and we explicitly state the trial phase and context (i.e., second-line setting). We have also included references to clinical benefit metrics and trial phase to contextualize the degree of “progress” or “promise” made. We modified the manuscript (see the new version added in red) as follow
Original text:
"...has shown compelling efficacy in heavily pretreated HER-2-positive GC patients..."
Modified according to the reviewer’s comment as follow :
...has shown notable clinical activity, including improvements in ORR and OS in heavily pretreated HER-2-positive GC patients, as demonstrated in DESTINY-Gastric01 and DESTINY-Gastric02 trials...
Original text: "...have demonstrated encouraging preclinical and clinical data..."
Modified as follow :
...have demonstrated encouraging preclinical and early clinical signals, particularly in first-line settings, although confirmatory evidence from large randomized trials is still awaited.
Original text :
"...have marked a significant shift in the treatment paradigm..."
Modified as follow :
...may represent a significant shift in the treatment paradigm, especially when combined with immunotherapy in selected patient subgroups...
Original text:
"...holds the promise of making CAR T-cell therapy a viable treatment option..."
Modified as follow :
...holds the potential, pending resolution of current technical and safety challenges, to make CAR T-cell therapy a viable treatment option...
Original text:
"...poised for significant progress." Modified as follow:
...poised for potentially meaningful advancements, provided ongoing clinical research can confirm efficacy in diverse patient populations.
Reviewer 2 Report
Comments and Suggestions for Authors
Major Concerns:
Targeting HER2 in advance studies is too descriptive; the authors should consider making a table to explain it in a simple version.
Minor Concerns;
Some of the abbreviations are repeated - eg. line 52, line 61, etc.
Comments on the Quality of English LanguageOverall, all languages can be improved to make them more straightforward for readers.
Author Response
Comment 1: Major Concerns: Targeting HER2 in advance studies is too descriptive; the authors should consider making a table to explain it in a simple version.
Answer: We sincerely thank the reviewer for his/her valuable comments and suggestions. We greatly appreciate the time and attention he/she have dedicated to our work. Regarding the request to summarize the data in the table, we'd prefer to maintain the current structure, considering that the manuscript already includes a very detailed table related to the perioperative and neoadjuvant phases, along with two illustrative figures. We believe that further summarization might excessively weigh down the manuscript and affect its clarity.
Comment 2: Some of the abbreviations are repeated - eg. line 52, line 61, etc
Answer: We modified the text, accordingly. See the part in red in the new version of the manuscript
Comment 3: Overall, all languages can be improved to make them more straightforward for readers
Answer: We sincerely thank you for your valuable suggestions. Regarding your comment on the English language, we would like to kindly inform the reviewer that the manuscript is always reviewed by our internal native English-speaking editor to ensure accuracy and fluency. Nevertheless, we remain fully open and willing to further revise the text should you consider it necessary.
Round 2
Reviewer 2 Report
Comments and Suggestions for Authors
The authors have addressed all my concerns.
Author Response
Comments 1: The authors have addressed all my concerns.
Response: thank you